# Burden of Healthcare Utilization among Chronic Obstructive Pulmonary Disease Patients with and without Cancer Receiving Palliative Care: A Population-Based Study in Taiwan

**DOI:** 10.3390/ijerph17144980

**Published:** 2020-07-10

**Authors:** Li-Ting Kao, Kuo-Chen Cheng, Chin-Ming Chen, Shian-Chin Ko, Ping-Jen Chen, Kuang-Ming Liao, Chung-Han Ho

**Affiliations:** 1Department of Respiratory Therapy, Chi Mei Medical Center, Tainan 71004, Taiwan; litingrt520@gmail.com; 2Department of Internal Medicine, Chi Mei Medical Center, Tainan 71004, Taiwan; kcg.cheng@gmail.com; 3Department of Safety Health and Environment, Chung Hwa University of Medical Technology, 71703 Tainan, Taiwan; 4Department of Intensive Care Medicine, Chi Mei Medical Center, Tainan 71004, Taiwan; chencm3383@gmail.com; 5Palliative Care Center, Chi-Mei Medical Center, Tainan 71004, Taiwan; 737005@mail.chimei.org.tw; 6Marie Curie Palliative Care Research Department, Division of Psychiatry, University College London, London W1T 7NF, UK; pingjen.chen@gmail.com; 7Department of Family Medicine and Division of Geriatrics and Gerontology, Kaohsiung Medical University Hospital, Kaohsiung Medical University, Kaohsiung 80756, Taiwan; 8School of Medicine, Kaohsiung Medical University, Kaohsiung 80756, Taiwan; 9Department of Internal Medicine, Chi Mei Medical Center, Chiali, Tainan 72263, Taiwan; 10Department of Medical Research, Chi Mei Medical Center, Tainan 71004, Taiwan; 11Department of Hospital and Health Care Management, Chia Nan University of Pharmacy and Science, Tainan 71710, Taiwan

**Keywords:** palliative care, chronic obstructive pulmonary disease, healthcare utilization

## Abstract

Chronic obstructive pulmonary disease (COPD) is a chronic disease that burdens patients worldwide. This study aims to discover the burdens of health services among COPD patients who received palliative care (PC). Study subjects were identified as COPD patients with ICU and PC records between 2009 and 2013 in Taiwan’s National Health Insurance Research Database. The burdens of healthcare utilization were analyzed using logistic regression to estimate the difference between those with and without cancer. Of all 1215 COPD patients receiving PC, patients without cancer were older and had more comorbidities, higher rates of ICU admissions, and longer ICU stays than those with cancer. COPD patients with cancer received significantly more blood transfusions (Odds Ratio, OR: 1.66; 95% C.I.: 1.11–2.49) and computed tomography scans (OR: 1.88; 95% C.I.: 1.10–3.22) compared with those without cancer. Bronchoscopic interventions (OR: 0.26; 95% C.I.: 0.07–0.97) and inpatient physical restraints (OR: 0.24; 95% C.I.: 0.08–0.72) were significantly more utilized in patients without cancer. COPD patients without cancer appeared to receive more invasive healthcare interventions than those without cancer. The unmet needs and preferences of patients in the life-limiting stage should be taken into consideration for the quality of care in the ICU environment.

## 1. Introduction

Chronic obstructive pulmonary disease (COPD), a chronic progressive illness with a high mortality rate, is a global burden to which the high utilization of healthcare resources is attributed [1]. COPD characteristics, such as persistent breathlessness, productive cough, and repeated exacerbations, may attenuate lung function and also increase the burdens of healthcare utilization, as well as treatment and control costs, while the exacerbations of COPD worsen [2,3,4]. An acute episode of breathlessness in COPD is life-threatening and usually requires immediate hospital admission or care in an intensive care unit (ICU) to relieve the perceived breathlessness of patients [5]. A cross-national study involving 14 countries showed that the majority of patients with COPD appeared to have a high possibility of dying in hospital when compared with those with lung cancer [6]. The in-hospital mortality rate for ICU COPD patients with acute exacerbation and endotracheal (ET) intubation is about 25% [7,8,9]. Additionally, over half of COPD patients have one or more chronic diseases, including congestive heart disease (CHF) and diabetes mellitus (DM), which may increase the frequency of visits to the emergency department (ED), the length of hospitalizations, and COPD-related costs [10,11,12,13]. Therefore, managing COPD patients may require integrated care to meet the multi-dimensional needs of COPD patients and deliver the appropriate healthcare resources to those patients [14,15,16].

Palliative care (PC) is a type of care for ameliorating severe, longstanding health problems of patients with diseases such as cancer, CHF, and COPD in the life-limiting stage. PC not only provides coordinated and cooperative care in line with the curative and restorative treatments to patients who share multi-symptomatic problems, such as pain, breathlessness, depression and anxiety, insomnia, and so on, but also helps patients’ caregivers to face their grief and bereavement after the patient’s death [14,17,18]. Studies on the benefits of PC in cancer patients who received early PC intervention have shown that these patients may have a better quality of life, fewer depressive symptoms, and fewer aggressive end-of-life interventions. Similarly, PC in COPD patients seemed to significantly decrease ICU mortality, invasive mechanical ventilation (MV) use, cardiopulmonary resuscitation (CPCR), and daily medical costs [19,20,21,22]. According to a 10 year observational study of COPD patients admitted to the ICU, ICU stays lengthened (from 21.58 to 23.14 days) and ICU mortality increased (from 14.97% to 30.98%) from 2003 to 2013, respectively [23]. Therefore, an incentive payment program may benefit the promotion of PC utilization in severely ill and frail patients in any setting, with the aim of improving the quality of care for those patients in the life-limiting stage.

However, a limited number of studies have examined the burdens of healthcare utilization in COPD patients with and without cancer who received inpatient PC. Our study aimed to discover and compare the differences in the burdens of healthcare utilization between patients with and without cancer by using a population database. We hypothesized that COPD patients without cancer were more likely than those with cancer to experience burdensome healthcare utilization in the life-limiting stage.

## 2. Materials and Methods 

### 2.1. Data Sources

Taiwan’s National Health Insurance Research Database (NHIRD) was used for the present study. The NHIRD claims data of the NHIRD were obtained from Taiwan’s single-payer National Health Insurance (NHI) program, which covers approximately 99.9% of Taiwan residents. All personal information in the NHIRD was de-identified and encrypted to ensure the confidentiality of personal information. Diagnoses in NHIRD are indicated by the International Classification of Diseases, Ninth Revision, Clinical Modification (ICD-9-CM) codes. In addition, the medical expenditure applications provide the prescription and procedure for each patient’s outpatient visits or hospitalization records. The study was approved by the Institutional Review Board at Chi-Mei Medical Center (IRB No. 10701-E01).

### 2.2. Availability of Data and Materials

The data that support the findings of this study are available from Taiwan’s National Health Insurance Research Database managed by Taiwan’s Bureau of National Health Insurance (NHI), Ministry of Health and Welfare, but access to these data is restricted; the information was used under license for the current study and is, thus, not publicly available. However, data are available from the authors upon reasonable request and with the permission of Taiwan’s Bureau of National Health Insurance (NHI), Ministry of Health and Welfare.

### 2.3. Study Design and Population

COPD patients who received PC between January 1, 2009, and December 31, 2012, were selected in this population-based cross-sectional study. In the years of 2000 and 2009, Taiwan’s NHI started to comprehensively cover expenditures for PC for the care of patients with cancer and eight advanced chronic diseases, respectively, with the aim of ensuring patients’ dignity and preferences before patient death. COPD was one of the eight advanced chronic diseases, along with heart diseases, deteriorating kidney and liver function, and neuromuscular degenerative diseases such as Parkinson’s disease and dementia. The criteria for providing PC intervention in patients with advanced COPD are (1) no improvement of symptomatic distresses; (2) severely deteriorated pulmonary function; (3) severe impairment of daily living function; (4) severe comorbidities; (5) more than two hospital admissions for acute exacerbation; (6) a history of acute respiratory failure; (7) home oxygen dependence; (8) the use of non-invasive mechanical ventilation [24].

Given the aforementioned criteria and the limitation of the claims data, the inclusion criteria for advanced COPD patients in this study were (1) aged 55 years or older, (2) diagnosed with COPD (diagnosis codes: ICD-9-CM: 490–492, 496), and (3) admitted to the ICU in the past six months. Therefore, participants were patients with advanced COPD who received PC after 1 January 2009. The definition of PC was based on medical expenditure applications. In addition, to avoid potential confounding effects, the following exclusion criteria were applied: (1) incomplete data related to the study variables; (2) patients diagnosed with asthma (ICD-9-CM: 493) or cystic fibrosis (ICD-9-CM: 277); (3) patients who received PC before January 1, 2009; (4) patients who received PC intervention for more than one year. Finally, the study population was divided into two groups—COPD without cancer (study group) and COPD with cancer (comparison group)—to assess the differences in the utilization of health resources between the two groups. Considering statistical power, patients with different types of cancer were incorporated into the reference group. 

### 2.4. Measurements

The variables of interest included demographic characteristics and the utilization of healthcare services one year prior to the inpatient admission date for PC. Demographic characteristics included age, gender, and the presence of comorbid conditions. Age was classified as (1) 55–64 years, (2) 65–74 years, (3) 75–85 years, or (4) ≧85 years. Gender was categorized as male or female. Comorbidity was dichotomously measured as the presence or absence of comorbid conditions. Comorbid diseases of COPD patients in this study were based on the 2013 Global Initiative for Chronic Obstructive Lung Disease (GOLD) [7]. Comorbidities were defined as three outpatient visits or one inpatient admission within one year before the COPD diagnosis. The comorbid conditions included dementia (ICD-9-CM: 290), diabetes (ICD-9-CM: 250), heart failure (HF, ICD-9-CM: 428), liver cirrhosis (ICD-9-CM: 571.2, 571.5, 571.6), renal failure (ICD-9-CM: 582, 583, 585, 586, 588), ischemia heart disease (IHD, ICD-9-CM: 410-414), stroke (ICD-9-CM: 430-438), atrial fibrillation (AF, ICD-9-CM: 427.31), hypertension (ICD-9-CM: 401-405), anxiety or depression (ICD-9-CM: 296.2, 296.3, 298.0, 300.4, 309.0, 309.1, 296.2, 296.3, 298.0, 300.4, 309.0, 309.1, 311, 300.00, 300.01, 300.02, 300.09, 300.21, 300.22, 300.23, 309.24, 309.81), pneumonia (ICD-9-CM: 486), pulmonary tuberculosis (ICD-9-CM: 011), and osteoporosis (ICD-9-CM: 733.0). 

The utilization of healthcare resources included the frequency of emergency department (ED) visits, number of hospitalizations and ICU admissions, clinical examinations, and medical care interventions (invasive/non-invasive). Clinical examinations included computed tomography (CT) scans and echoes. Invasive medical care interventions included enteral tube insertion, tube feeding, blood transfusion, bronchoscope, panendoscope, mechanical ventilation (MV), endotracheal intubation (ET), tracheotomy, physical restraints, hemodialysis, defibrillation and temporary pacemaker, and cardiopulmonary resuscitation (CPCR). The non-invasive medical care intervention was non-invasive MV. Medical expenditure applications were used to identify clinical examinations and medical care interventions. Furthermore, the length of time to death among COPD patients receiving PC was also considered to avoid potential duration bias for some patients who had a longer end-of-life period.

### 2.5. Statistical Analysis

All statistical analyses were performed using Statistical Analysis System (SAS) (version 9.4; SAS Institute, Inc., Cary, NC, USA). The difference between COPD patients with and without cancer who received PC was assessed using Pearson’s chi-square test for categorical variables and Student’s t-test or Wilcoxon rank-sum test for continuous variables according to the results of the normality test. Linear regression analysis was used to estimate the adjusted correlation between study groups and the ED visits, hospitalizations, and ICU admissions. The association between each study group and the burden of healthcare utilization was calculated using a logistic regression model after adjusting for the confounding factors. All significance levels were set at *p*-value < 0.05.

## 3. Results

A total of 1215 COPD patients who received PC were divided into two groups—COPD with cancer (N = 816, 67.16%) and COPD without cancer (N = 399, 32.84%). The characteristics of the two groups are presented in Table 1. The majority of COPD patients were male. The mean age of COPD patients without cancer was significantly older than those with cancer (80.90 ± 8.53 vs. 75.63 ± 9.60 years; *p* < 0.0001). Patients without cancer who were aged >75 years accounted for 75.94% of the sample, which was more than the proportion of patients with cancer (55.14%, *p* < 0.0001). COPD patients without cancer significantly had coexisting dementia, HF, renal failure, IHD, AF, and osteoporosis (*p* < 0.05). The survival time of COPD patients without cancer was significantly longer than those with cancer (median (interquartile range, IQR): 1.2(0.60–2.40) vs. 0.96(0.36–2.04); *p* = 0.0232). The one year mortality rate from the index date of ICU admission of the two groups was approximately 80%.

Table 2 presents ED utilization, hospitalization, and ICU admission of patients in the previous one year for both groups. The rate and frequency of ED visits of patients did not show a significant difference between the two groups. Patients with and without cancer had a similar number and cumulative length of hospitalizations. Patients without cancer were more likely than those with cancer to have a higher mean ICU admission rate (10.65 ± 15.43 vs. 5.87 ± 6.64; *p* < 0.0001) as well as have a longer length of ICU stay (median (IQR): 34(15–90) vs. 15(6–36); *p* < 0.0001). After controlling for age, gender, and comorbidities, COPD patients without cancer also had an increase in ICU admission times (*Beta-coefficient*: 0.16; 95% C.I.: (0.11–0.22); *p* < 0.0001) and the length of ICU stay (*Beta-coefficient*: 0.22; 95% C.I.: (0.16–0.28); *p* < 0.0001), compared to those with cancer. 

Figure 1 presents the distribution of coexisting diseases of COPD patients without cancer. The most common comorbidities were HF (34.09%), end-stage neurological diseases (31.0%), and chronic hepatic disease (24.32%). Figure 2 presents the proportions of cancer types (based on the categorizations of ICD-9-CM) of COPD patients with cancer. The most common types of cancer were in the respiratory and intrathoracic organs (35.78%), digestive organ and peritoneum (33.33%), and genitourinary organs (12.13%). Lung cancer, colon cancer and rectal cancer, and liver cancer were greatly accounted for (33.46%, 14.22%, and 8.33%, respectively) among all cancer diseases.

Table 3 shows the differences in healthcare utilization burdens between patients with and without cancer. Patients with cancer appeared to receive significantly more blood transfusions than those without cancer (OR: 1.66; 95% C.I.: (1.11–2.49); *p* = 0.0124). Patients with cancer who underwent CT scans showed a statistical difference from those without cancer (OR: 1.88; 95% C.I.: (1.10–3.22); *p* = 0.0225). Bronchoscopy (OR: 0.26; 95% C.I.: (0.07–0.97); *p* = 0.0472) and inpatient physical restraints (OR: 0.24; 95% C.I.: (0.08–0.72); *p* = 0.0153) were more likely to be utilized in patients without cancer than those with cancer.

## 4. Discussion

This retrospective cross-sectional study aimed to discover and compare the different burdens of healthcare utilization between COPD patients with and without cancer who received PC. Patients without cancer were older and had a high frequency of dementia, HF, renal failure, IHD, and AF. Compared with patients with cancer, those without cancer appeared to have a higher frequency of ICU admissions and a longer ICU stay as well as receive more bronchoscopic interventions and inpatient physical restraints. On the other hand, patients with cancer received more blood transfusions and CT scans than those without cancer. The data indicate that approximately 80% of severely ill COPD patients with and without cancer who received inpatient PC died during the one year follow-up after the date of ICU admission. Therefore, we suggest that practitioners in real clinical practice who care for severely ill COPD patients with and without cancer take those differences into consideration to examine whether the utilization of healthcare resources could potentially burden vulnerable patients.

The burdens of healthcare utilization appeared to be affected by cancer among COPD patients with and without cancer, similar to previous study findings for patients with COPD and lung cancer [25,26]. Our study found that invasive interventions, such as enteral tube insertion, invasive and non-invasive MV, ET intubation, tracheotomy, hemodialysis, and CPCR, were highly utilized in patients without cancer compared with patients with cancer. Additionally, patients without cancer had significantly longer hospitalization and ICU stays and significantly more ICU admissions than those with cancer. The aforementioned healthcare resource utilization is an attempt to prolong the lives of patients without cancer, and it presents a dilemma in caring for COPD patients with unpredictable prognoses and lifespans. Previous studies presented similar findings, reporting that COPD patients were more likely to receive healthcare resources before their death than lung cancer patients [27,28]. As a result, patients with COPD might undergo additional suffering that is attributed not only to the disease itself but also to the iatrogenic procedures of standard medical care [27,28]. Therefore, to provide better ICU care to patients with severe chronic illnesses such as COPD, a shared-decision model (SDM) may be useful to facilitate open communication and discussion with patients and their families in order to achieve the goals of high quality care, such as reducing aggressive medical interventions, earlier discharge from the ICU, and lower death rates in the hospital in the end-of-life stage [29,30,31,32].

The use of inpatient physical restraints for severely ill patients seems to reflect a pattern of care that does not meet patient needs. Few studies have discussed the use of physical restraints in end-of-life healthcare utilization for patients with COPD and lung cancer or COPD patients without cancer. Physical restraints are used as a safety measure to limit physical activities and may decrease the risk of self-harm, which is usually attributed to a response related to human defense instincts. According to previous studies, physical restraints are considered as an approach to protect severely ill patients from unintentionally pulling out medical intervention equipment, which is typical in the ICU environment [33,34]. Factors that have been associated with the use of physical restraints in patients in ICUs were older age, MV use, and a low patient-to-nurse ratio [35,36]. Unfortunately, 16.3% of deceased patients had experienced physical restraints by the time of death, and approximately one out of four ICU patients died in restraints [37]. Our study may be the first to explore the utilization of inpatient physical restraints in the care of ICU COPD patients with and without cancer who received inpatient PC and also point out the difference between them. The results of our study highly suggest that clinical care did not meet the needs and preferences of the COPD patients who did not have cancer, even though they received inpatient PC. It may be worthwhile for healthcare professionals to take the findings of this study into consideration to find solutions such as chemical restraints or other alternatives to physical restraints in an attempt to improve the quality of holistic care for patients in the life-limiting condition and also avoid imposing inappropriate medical care on them.

The use of non-invasive or invasive MV in the care of severely ill patients appears to depend on the goal of care in the life-limiting stage [20,21,27,28]. Our study found that non-invasive and invasive MV treatment was applied for COPD patients without cancer more often than for those with cancer. Several studies have similarly shown greater non-invasive MV utilization in patients with COPD compared with those with lung cancer [20,21,27,28]. In essence, non-invasive MV treatment plays a critical role in COPD patients with acute/acute-on-chronic respiratory failure in the process of medical care during hospitalizations. According to the official European Respiratory Society (ERS) and American Thoracic Society (ATS) clinical practice guidelines in 2017, the utilization of non-invasive MV is strongly recommended for patients with acute respiratory failure (ARF) who suffer from hypercapnia due to COPD exacerbation and for patients with cardiogenic pulmonary edema. It is also recommended for patients in palliative care as a ceiling of therapy for dyspnea. This evidence-based guideline indicates that non-invasive MV is more likely to not only decrease the need for endotracheal intubation and mechanical ventilation for exacerbated COPD patients with hypercapnia ARF, but also increase hospital survival and quality of life for patients with COPD and congestive heart failure with the status of “do not intubate” [38]. The purposeful intervention of non-invasive MV in COPD patients could be an alternative to first-line ET intubation and may preserve the speaking capacity of patients, allowing them to communicate with and convey their wills to their families before they die. However, when deciding on the use of non-invasive MV treatment as a form of care, medical care professionals should consider its advantages and disadvantages; this treatment can involve several discomforts related to the mask interface, such as the tightening of facial skin and facial stress ulcers, claustrophobia, stomach distension, and so on. Our study results reveal the difference in the quality of PC between COPD with and without cancer in the ICU care environment, and they are likely to unveil limitations in the understanding of clinicians, patients, and family caregivers about the inner merit of PC in severe COPD patients without cancer, regardless of whether these patients are older, have additional coexisting chronic diseases, or have more ICU admissions and ICU stays. Therefore, the early integration of professional caregivers may be of benefit by improving the use and timing of PC in any setting, allowing for the timely provision of an individualized plan of palliative care appropriate for the disease severity of these patients and further reducing the utilization of medical care resources [39,40,41].

The strength of this study lies in the large sample size and the generalizability of results by examining patients across all of Taiwan. Using the population database, we were able to investigate the different utilizations of healthcare resources between COPD patients with and without cancer who received inpatient PC after a one year follow-up of the date of ICU admission. This study has some limitations. A major limitation is that the disease severity and functional status in the claims database may have confounded our results. However, as this study focused on COPD patients in the ICU, we believe that the confounding effects of disease severity and functional status could be reduced. In addition, the exact initial date of patients who received PC could not be identified, and therefore, the hospital admission date was used as the date of receiving PC. The potential overestimated duration of time to death may have affected the calculated frequency of health utilization of palliative care services and life-sustaining treatment, but the bias could be decreased by using an adjustment approach to estimate the association effects. Moreover, the limitations of misclassification of the diagnoses coded in the dataset could have contributed to an underestimation of comorbidities. However, the misclassification bias of NHIRD has been validated by previous studies [42,43]. Finally, a potential secondary data bias of old claims data for this study could lead to an incomplete depiction of the current situation among COPD patients receiving PC. In future research, it necessary to obtain up-to-date data and collect more study subjects for a longitudinal claims cohort.

## 5. Conclusions

Disparities in healthcare resources remain among COPD patients with and without cancer receiving inpatient PC. ICU patients without cancer are more burdened by healthcare resource utilization than those with cancer. From the view of holistic care, the inner merit of PC is equally applied to the care of all patients with severe chronic diseases such as cancer and COPD throughout all stages of disease severity. However, patients with such severe cases of COPD without cancer usually receive more aggressive medical care in an attempt to cure and restore the severely problematic illness. The goal of treatment in patients without cancer might cause these patients and their caregivers to bear more physical and psychological burdens before the patient’s death. The findings of this study highlight the fragmented goal of PC in caring for COPD without cancer and also reflect a lack of positive attitude in professional caregivers in the face of prognostication difficulties. Further research can use this study to develop a PC training course about COPD for programming individualized and/or patient-led SDMs and advanced care planning (ACP) in any medical setting, community, or school. Measures that are proactively integrated with public health education, professional proclamation, and charity events related to COPD could perhaps urge the general population to arouse and improve the awareness of PC in COPD.

## Figures and Tables

**Figure 1 ijerph-17-04980-f001:**
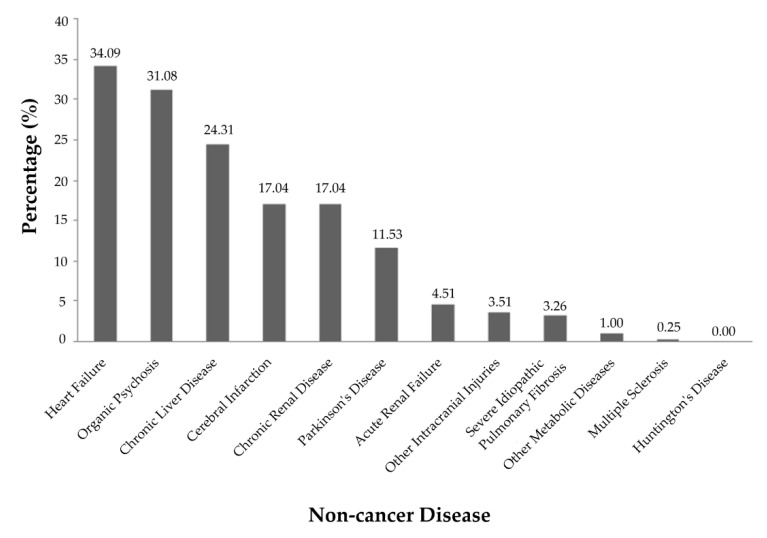
The distribution of diseases among COPD patients without cancer who received PC.

**Figure 2 ijerph-17-04980-f002:**
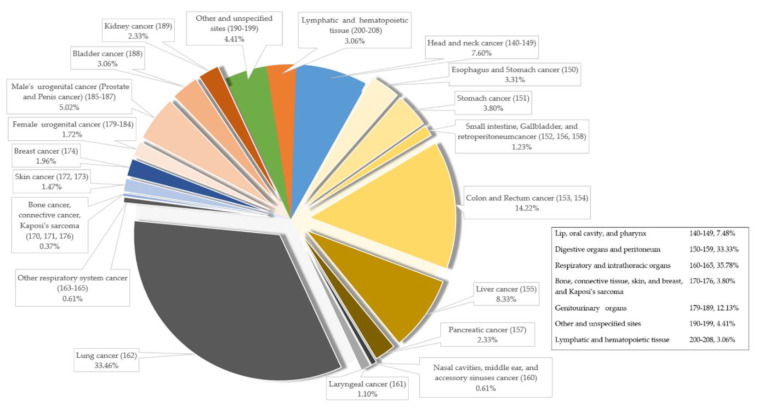
The distribution of cancer types among COPD patients who received PC.

**Table 1 ijerph-17-04980-t001:** Demographic characteristics of COPD patients with and without cancer receiving PC.

Variable	Cancer Patients (N = 816)	Non-Cancer (N = 399)	*p*-Value
**Age, mean ± SD**	75.63 ± 9.60	80.90 ± 8.53	<0.0001
**Age Group, n(%)**			
**55–64**	138 (16.91)	24 (6.02)	<0.0001
**65–74**	228 (27.94)	72 (18.05)	
**75–84**	315 (38.60)	170 (42.61)	
**≧85**	135 (16.54)	133 (33.33)	
**Gender, n(%)**			
**Male**	632 (77.45)	269 (67.42)	0.0002
**Female**	184 (22.55)	130 (32.58)	
**Comorbidities, n(%)**			
**Dementia**	56 (6.86)	93 (23.31)	<0.0001
**Diabetes**	240 (29.41)	135 (33.83)	0.1283
**Heart failure**	102 (12.50)	106 (26.57)	<0.0001
**Liver cirrhosis**	78 (9.56)	29 (7.27)	0.1974
**Renal failure**	76 (9.31)	70 (17.54)	<0.0001
**IHD**	182 (22.30)	113 (28.32)	0.0228
**Stroke**	43 (5.27)	29 (7.27)	0.1953
**Atrial fibrillation**	54 (6.62)	43 (10.78)	0.0175
**Hypertension**	379 (46.45)	173 (43.36)	0.3265
**Anxiety and Depression**	45 (5.51)	15 (3.76)	0.2063
**Pneumonia**	369 (45.22)	187 (46.87)	0.6238
**Pulmonary tuberculosis**	33 (4.04)	15 (3.76)	0.8764
**Osteoporosis**	23 (2.82)	26 (6.52)	0.0030
**Death, n(%)**			
**Yes**	645 (79.04)	335 (83.96)	0.0444
**No**	171 (20.96)	64 (16.04)	
**Time to Death, month**			
**Median (IQR)**	0.96 (0.36–2.04)	1.20 (0.60–2.40)	0.0232

**Table 2 ijerph-17-04980-t002:** The utilization of healthcare resources for the frequency of ED visits, the number of hospitalizations, and the number of ICU admissions in COPD patients with and without cancer within one year before the inpatient date of PC.

Variable	Cancer Patients (N = 816)	Non-Cancer (N = 399)	*p*-Value	*Beta-Coefficient* * (95% CI)
**Patients ever visited ED in one year**				
Frequency of whether had ED visits, n(%)	719 (88.11)	357 (89.47)	0.5035	0.01 (−0.03–0.05)
Whether had ED visits, mean ± SD	3.73 ± 3.65	3.58 ± 3.59	0.5163	−0.05 (−0.12–0.01)
**Patients ever hospitalized in one year**				
The number of hospitalizations, mean ± SD	5.17 ± 3.36	5.19 ± 4.29	0.9468	−0.03 (−0.09–0.03)
The cumulative length of stay (days), median (IQR)	57 (35–87)	60 (35–99)	0.2555	0.01 (−0.05–0.07)
**ICU ever admitted in one year**				
The number of ICU admissions, mean ± SD	5.87 ± 6.64	10.65 ± 15.43	<0.0001	0.16 (0.11–0.22) ^#^
The cumulative length of ICU stay (days), median (IQR)	15 (6–36)	34 (15–90)	<0.0001	0.22 (0.16–0.28) ^#^

* The *Beta-coefficient* was adjusted by age, gender, and comorbidities; ^#^
*p* < 0.05.

**Table 3 ijerph-17-04980-t003:** The utilization of medical healthcare resources by COPD patients with and without cancer receiving PC during the hospitalization.

Variables	Cancer Patients (N = 816)	Non-Cancer (N = 399)	*p*-Value	Crude OR (95% CI)	Adjusted OR * (95% CI)
**Invasive life support**					
Blood transfusion (whole blood or red blood cells)	125 (15.32)	40 (10.03)	0.0124	1.62 (1.11–2.37) ^#^	1.66 (1.11–2.49) ^#^
Enteral tube insertion	166 (20.34)	83 (20.80)	0.8798	0.97 (0.72–1.31)	1.05 (0.76–1.45)
Tube feeding	186 (22.79)	80 (20.05)	0.3013	1.18 (0.88–1.58)	1.22 (0.88–1.68)
Invasive mechanical ventilation	50 (6.13)	31 (7.77)	0.3271	0.78 (0.49–1.23)	0.64 (0.39–1.06)
Non-invasive mechanical ventilation	29 (3.55)	22 (5.51)	0.1273	0.63 (0.36–1.11)	0.79 (0.42–1.47)
Endotracheal intubation	27 (3.31)	15 (3.76)	0.7385	0.88 (0.46–1.67)	0.85 (0.42–1.70)
Tracheostomy	2 (0.25)	3 (0.75)	0.3383	0.32 (0.05–1.95)	0.41 (0.06–2.85)
Hemodialysis	4 (0.49)	3 (0.75)	0.6898	0.65 (0.15–2.92)	1.13 (0.20–6.39)
Echo-guided intervention	14 (1.72)	4 (1.00)	0.4511	1.72 (0.56–5.27)	1.45 (0.44–4.81)
Bronchoscope	4 (0.49)	7 (1.75)	0.0472	0.27 (0.08–0.95) ^#^	0.26 (0.07–0.97) ^#^
Cardiopulmonary resuscitation (per 10 min)	17 (2.08)	9 (2.26)	0.8351	0.92 (0.41–2.09)	0.69 (0.28–1.65)
**Invasive examination**					
Abdominal echo	45 (5.51)	16 (4.01)	0.3273	1.40 (0.78–2.50)	1.59 (0.85–2.97)
Computed tomography	74 (9.07)	21 (5.26)	0.0225	1.80 (1.09–2.96) ^#^	1.88 (1.10–3.22) ^#^
**Invasive healthcare**					
Inpatient physical restraints	6 (0.74)	10 (2.51)	0.0153	0.29 (0.10–0.80) ^#^	0.24 (0.08–0.72) ^#^

* The OR was adjusted by age, gender, and comorbidities; ^#^
*p* < 0.05.

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
