# Peer review of "Burden of Healthcare Utilization among Chronic Obstructive Pulmonary Disease Patients with and without Cancer Receiving Palliative Care: A Population-Based Study in Taiwan"

_ijerph, 2020, doi:10.3390/ijerph17144980_

Round 1
Reviewer 1 Report
The manuscript compared the healthcare resources burden caused by COPD patients with and without cancer receiving PC, which could have a guiding significance to develop more efficient medical and clinical management for the COPD patients. Overall, the manuscript is well-organized in a logic manner, and characterized the difference of health care utilization by COPD patients with or without cancer based on the current Taiwan's National Health Insurance Research Database. I recommend its publication in International Journal of Environmental Research and Public Health.
Minor point:
1. I didn't see the figure and tables in the manuscript. Were these data compiled into the manuscript?
Reviewer 2 Report
Dear Authors,
Thank you for the opportunity to read your article. In my opinion it is a well prepared study and manuscript.
I have to points that I think would improve the quality of paper:
- conclusions - I think you should pay more emphasis on implications of your study, be more general showing what the results you obtain from analysis may bring to the broadly understand medical practice and public health;
- I would suggest to add few more up-to-date references, as I see many of positions are older than 5 years.
Author Response
Please see the attached PDF document for the response to reviewer's comments:

Reviewer 3 Report
The manuscript presented by Kao et al. is an analysis on the health care burden of COPD patients with and without lung cancer in Taiwan. In general, the quality of the presentation and the scientific soundness are acceptable. The topic is of high interest and significant for the reader. Yet, data is 10 years old and some issues arise concerning the results. After exclusion of material and methods, results and references, an electronic check for plagiarism resulted in a value of 2,8% (an acceptable result from the perspective of the reviewer). The reviewer has the following concerns:
1) It does not become entirely clear what comparision was performed. It line 115, it is stated: "...divided into two groups, COPD without cancer (study group) and COPD with lung cancer (comparison group). Yet, in the following and according to Figure 2, patients suffered from cancer of various origin. Further, "lung cancer" is not sufficiently precise. The reviewer would suggest to clearly state which diagnoses of (lung? other?) cancer were included in the study. Further, it is not clear whether there are differences between different types of lung cancer regarding the results. The authors should at least comment on that.
2) Major concern: Why was data only selected from the years 2009 until 2012? Although the reviewer understands that data collection and analysis sometimes takes some time, this data is over 10 years (!) old. Is newer data not available? The authors should explain and need to comment on this, and mention this fact and potential changes over the last years at least as a limitation in the discussion section.
3) English grammer and style have to be improved significantly.
Author Response

(The authors gave the same response as above.)

Round 2
Reviewer 3 Report
The reviewer has no further concerns. The authors have responded adequately to the points of the reviewer.
Author Response
We thank the reviewer’s helpful comments.